# Behavioural issues in late life may be the precursor of dementia- A cross sectional evidence from memory clinic of AIIMS, India

Abhijith Rajaram Rao[1]ᵒ, Prasun Chatterjee🅞[1]ᵒ*, Meenal Thakral[1], S. N. Dwivedi[2], Aparajit Ballav Dey[1]

1 Department of Geriatric medicine, All India Institute of Medical Science, New Delhi, India, 2 Department of Biostatistics, All India Institute of Medical Science, New Delhi, India

ᵒ These authors contributed equally to this work.
* drprasun.geriatrics@gmail.com

## Abstract

**Data Availability Statement:** All relevant data are within the manuscript and its Supporting Information files.

### Background

Mild Behavioural Impairment (MBI), an "at risk" state for incident cognitive declin, is characterized by late onset, sustained neuropsychiatric symptoms of any severity which cannot be accounted for by other formal medical and psychiatric nosology. There is no study related to MBI from India.

### Methods and findings

In this cross-sectional observational study 124 subjects 60 years and above were recruited between March 2017 to October 2018, from memory clinic of department of Geriatric medicine with memory or behavioural complains. Subjects with major neurocognitive impairment (CDR score of 1 or more), major depressive disorder, generalized anxiety disorder and impaired activities of daily living (ADL) were excluded. Subjects with Mild Cognitive impairment (MCI) (CDR- 0.5), and Subjective cognitive impairment (SCI) (CDR- 0) were included. Neuropsychiatric Inventory Questionnaire (NPI-Q) was used to identify the presence of NPS. The ISTAART-MBI (International Society of Advance Alzheimer's Research and Treatment-Alzheimer's Association) diagnostic criteria was used to diagnose MBI. All the participants underwent a geriatric assessment using standardised screening. The objectives of this study was to determine the frequency of mild behavioural impairment (MBI), and its domains, in MCI or SCI and its association with comorbidities and geriatric syndromes. The mean age of the participants was 69.21, 71.77% (89) were male and 28.23% (35) were female. 41.13% (51) of these individuals were diagnosed with MBI. The MBI and non MBI group differed significantly in marital status, cognitive status and MCI subtype. The proportion of domains involved are as follows: decreased motivation 60.78%(31), emotional dysregulation 54.90% (28), impulse dyscontrol 68.63% (35), social inappropriateness 21.57% (11), abnormal perception 2 (3.93%). Presence of multi-morbidity, and diabetes, were statistically significant between the groups.

**Funding:** The authors received no specific funding for this work.

**Competing interests:** The authors have declared that no competing interests exist.

## Conclusion

This study presents the first clinic-based prevalence estimates of MBI from Asia. Findings indicate a relatively high prevalence of MBI in predementia clinical states, impulse dyscontrol was the most commonly involved MBI domain. Multimorbidity, diabetes, urinary incontinence were other determinants of MBI.

## Introduction

Mild Behavioural Impairment (MBI) is identified as an "at-risk" state for incident cognitive decline and dementia. It might be the index manifestation of neurodegeneration in a substantial number of individuals who would eventually progress to develop dementia[1]. MBI is characterised by late-onset, impactful and sustained neuropsychiatric symptoms (NPS) of any severity which cannot be accounted for by other formal medical and psychiatric nosology. MBI distinguishes between formal psychiatric illness and chronic psychiatric symptomatology, from new-onset psychiatric symptoms in older adults. The MBI diagnostic criteria was proposed by the International Society of Advance Alzheimer's Research and Treatment–Alzheimer's Association, (ISTAART-AT). It classifies MBI into five domains which include decreased motivation or drive, emotional/affective dysregulation, impulse dyscontrol and agitation, social inappropriateness, and delusions and hallucinations.

There is increasing evidence which links the presence of NPS prior to the onset of cognitive symptoms to a greater likelihood of cognitive impairment[2]. NPS presents as the first indicator of impending dementia in many individuals. In the past, the early presentation of NPS in the course of the neurodegenerative disease would make the clinician suspect behavioural variant Frontotemporal Dementia (bvFTD), but Tarango et al. [3] observed that NPS could emerge in advance of any dementia syndrome. In a prospective observational study of nondemented individuals with NPS, it was seen that 36% of individuals developed FTD, 28% developed AD, 18% progressed to Vascular Dementia (VaD) and 18% other types dementia. A link exists between NPS in MCI and non-MCI[4–6] and increased risk of progressing to dementia, and MBI has been found to be common with the prevalence ranging between 14.15–81.5%[7,8].

It is undeniable that dementia would be the future epidemic of ongoing demographic transition in India[9]. So there is a dire need for identification of pre-dementia state and their risk factors which intern will allow for primary and secondary prevention of dementia. So, we intended to find the epidemiology of MBI in the health-seeking older adults and its determinants.

## Materials and methods

### Participants

A written informed consent was taken from all participants. Subjects attending memory clinic with the complaint of memory impairment, or behavioural changes were recruited. This clinic is held on every Wednesday, in the department of Geriatric medicine, in a tertiary health centre with the purpose of detailed evaluation of patients who are referred for assessment of cognitive decline. It caters to people from New Delhi as well as nearby cities and states. The clinic is manned by the multidisciplinary team, which includes, Geriatrician, neuropsychologist, occupational therapist, physiotherapist and dietician, to provide comprehensive assessment and care for older adults with cognitive decline. This study was conducted from January 2017 till October 2018. Of the total 366 new subjects who attended memory clinic, 124 subjects were included. Clinical Dementia Rating (CDR) scale was used to assess cognitive status. A certified

CDR rater conducted the interview. Subjects with dementia, as diagnosed by a CDR score of 1 or more, and impaired activities of daily living (ADLs) were excluded. Subjects with a CDR score of 0.5 were diagnosed as Mild cognitive impairment (MCI), and those with a CDR score of 0 were diagnosed as Subjective cognitive impairment (SCI).

## Measures

Neuropsychiatric Inventory Questionnaire (NPI-Q) was used to identify the presence of Neuropsychiatric Symptoms (NPS). The form was completed by a psychologist with five years of experience in assessing dementia patients.

To identify the five ISTAART-AA MBI domains, ten NPS domains from the NPI-Q were used. Decreased motivation was identified by NPI-Q apathy; emotional dysregulation by the NPI-Q depression, anxiety, elation; impulse dyscontrol by NPI-Q agitation, irritability, motor disturbance; social inappropriateness by NPI-Q disinhibition; and abnormal perception by NPI-Q delusions, hallucinations. It was indicated as "yes" if the symptoms had been present for at least past six months. 15 item Geriatric depression scale (GDS) and 7 item generalized anxiety disorder (GAD-7) were used to identify depression and anxiety.

The ISTAART-MBI diagnostic criteria was used to diagnose MBI. To determine criterion one NPI-Q was administered, and six-month symptom duration was enquired. For criterion two, information was obtained from the informant. For criterion three Mini International Neuropsychiatric Interview (MINI) version 5.0.0 was used to diagnose psychiatric disorders like generalised anxiety disorder, major depression, manic or psychotic disorders.

Various geriatric syndromes like vision, hearing, mobility/risk of fall, urinary incontinence, and nutrition were assessed using standardised screening tool.

The vision was assessed by a screening question "Do you have any difficulty in seeing a car from long distance or reading or difficulty in doing any of your daily activities because of your eyesight?" The replay was considered as positive if the subject replied as yes, and negative if the answer was no. Hearing was assessed by the whisper test. It was administered by first standing behind the subject to prevent speech reading, the subject was asked to cover one ear, and then three random words (with two-syllable) were whispered at a distance of 12 inches from the person's ear and then the subject was asked to repeat the words. Inability to repeat all the words in both ears or either ear was considered as positive. Leg mobility and risk of fall was assessed by the time taken by the subject in seconds to rise from an armchair, walk a distance of three metres with no obstacles at a comfortable pace, turn around and return to the chair and sit. Total time of more than equal to 13.5 seconds taken to complete this task was considered as positive. Urinary incontinence was assessed by asking two questions "In the last year, have you ever lost your urine and gotten wet?" If the subject says yes, the next question was "Have you lost urine on at least six separate days?" If the subject replies yes to both the questions, it was considered as positive. Nutrition/weight loss was assessed by asking "Have you lost 5% of your total body weight over the past six months without trying to do so?".

Participants were enquired about various co-morbidities, including the presence of hypertension, diabetes, hypothyroidism, coronary artery disease, cerebrovascular accident, chronic obstructive airway disease and hyperlipidaemia.

## Statistical analysis

Data were analysed using STATA 13. Categorical variables were described in terms of frequency and percentage. Non-categorical variables were described as mean, median, standard deviation, quartiles. Association between two categorical variables were assessed using the Pearson Chi-square test/Fisher exact test. To compare quantitative variables between MBI and

non-MBI unpaired t-test/Wilcoxon rank-sum test were used. The level of significance was set at p<0.05.

## Results

Sample characteristics are presented in Table 1 (n = 124). Among the study subjects, MCI was diagnosed in 68.54%, and SCI was diagnosed in 31.45%. Neuropsychiatric symptoms were present in 55.65% of the total participants, and 51 (41.12%) had the diagnosis of MBI. The MBI and non-MBI groups differed significantly (p<0.05) in marital status(p = 0.024), cognitive status (0.048) and MCI subtype (0.030). Statistically significant difference was noted in GDS, NPI-Q and MBI Checklist total scores between the groups. There was no significant difference with gender, age, literacy status, socioeconomic status. The mean age in the MBI and non-MBI groups were comparable (68.5 vs 69.7 years), 29.33% of men and 45.71% of women had MBI. The participants in both the group belonged to upper middle (33.33% vs 43.83%), lower middle (33.33% vs 36.98%) and upper lower (31.37% vs 19.17%) socioeconomic status in the Kuppuswamy scale. The median years of education was 10 years in both the group, 13% had no formal education. The smoking and alcohol consumption history were similar between the groups. In MBI and non-MBI froup, 96% and 95% had memory impairment as their chief complaints. 64.7% participants or their relative had behavioural changes as their chief complain in the MBI group as compared to 15.06% in non-MBI group. The median score of Geriatric depression scale was 4 in MBI group and 3 in non-MBI group. The NPI-Q score was 4 and 0 and the MBI checklist score was 7 and 0 respectively.

The most common neuropsychiatric symptoms noted was Impulse dyscontrol (68.63%), followed by decreased motivation (60.78%) and emotional dysregulation (54.90%) (Table 2). Social inappropriateness and abnormal perception were relatively uncommon (21.57% and 3.92% respectively).

Amongst the Geriatric syndromes (Table 3) and comorbidity (Table 4), urinary incontinence (p = 0.001) and diabetes was significantly higher in the MBI subjects (Table 3). Visual impairment (47.06% vs 42.47%) and hearing impairment (31.37% vs 31.51%) were similar between groups. The mean time taken for TUG was 13.25 and 12.80 seconds (p = 0.252). Risk of malnutrition, though was different, it was not statistically significant (15.69% vs 8.22%; p = 0.196). 58.82% of participants with MBI had two or more co-morbidities when compared to 35.62% participants without MBI (p = 0.037). Hypertension (64.7% vs 56.16%), hypothyroidism (13.73% vs 6.85%), CAD (11.76 vs 63.83%), CVA (9.80% vs 6.85%), COPD (9.80% vs 10.96%) and hyperlipidemia (5.88% vs 5.48%) were not significantly different between the groups. Figure in S1 Fig.

## Discussion

This study is the first to determine the epidemiology of MBI in a memory clinic setting from Asia. The findings stressed the fact that MBI is highly prevalent (41.12%) in subjects with amnestic MCI, multimorbidity had a significant association with MBI and impulse dyscontrol was the most commonly involved neuropsycholofical symotpm.

It was noted that the prevalence of MBI, using the ISTAART-AA MBI diagnostic criteria, in subjects with MCI and SCI in memory clinic setting was 47.06% and 28.20% respectively. Interestingly the frequency of MBI was increasing with cognitive decline in the study participants. A study with a similar methodology in memory clinic found the prevalence of MBI to be 81.5%[7], whereas it was 3.5% in a psychiatric outpatient clinic[10]. This difference could be due to differences in demography, study settings and instruments selected to identify NPS. The NPI-Q requires one month of symptoms as the reference frame. But for this study, we

**Table 1. Descriptive parameter of the participants.**

| Variables | MBI (n = 51) | Non-MBI (n = 73) | p-value |
|---|---|---|---|
| | Frequency (%) | Frequency (%) | |
| **Age** | | | |
| <75 | 43 (84.31%) | 57 (78.08%) | 0.387 |
| ≥75 | 8 (15.69%) | 16 (21.92%) | |
| Mean age ± SD | 68.50 ± 6.66 | 69.71 ± 6.64 | 0.323 |
| **Gender** | | | |
| Male | 35 (39.33%) | 54 (73.97%) | 0.515 |
| Female | 16 (45.71%) | 19 (26.02%) | |
| **Body mass index (BMI) [kg/m$^2$]** | | | |
| <18.5 | 4 (7.84%) | 4 (5.47%) | 0.398 |
| 18.5–22.9 | 26 (50.98%) | 30 (41.09%) | |
| ≥23 | 21 (41.17%) | 39 (53.42%) | |
| **Socioeconomic status** | | | |
| Upper | 1 (1.96%) | 0 (0.00%) | 0.238 |
| Upper Middle | 17 (33.33%) | 32 (43.83%) | |
| Lower Middle | 17 (33.33%) | 27 (36.98%) | |
| Upper Lower | 16 (31.37%) | 14 (19.17%) | |
| **Literacy status** | | | |
| No formal Education | 7 (13.73%) | 10 (13.70%) | 0.505 |
| I to X | 26 (50.98%) | 30 (41.10%) | |
| X and above | 18 (35.29%) | 33 (45.21%) | |
| **Years of Education** | | | |
| Median | 10 | 10 | 0.533 |
| Range | 0–20 | 0–18 | |
| **Marital Status** | | | |
| Married | 47 (92.15%) | 56 (76.71%) | **0.024** |
| Widow/Widower | 4 (7.85%) | 17 (23.28%) | |
| **Retired/still working** | | | |
| Retired | 23 (45.09%) | 35 (47.94%) | 0.870 |
| Still working | 14 (27.45%) | 21 (28.76%) | |
| Home maker | 14 (27.45%) | 17 (23.28%) | |
| **History of smoking** | | | |
| Current | 7 (13.72%) | 7 (9.58%) | 0.562 |
| Ex-smoker | 13 (25.49%) | 15 (20.54%) | |
| Never | 31 (60.78%) | 51 (69.86%) | |
| **History of Alcohol** | | | |
| Yes | 9 (17.64%) | 11 (15.06%) | 0.393 |
| No | 42 (82.35%) | 62 (84.93%) | |
| **Chief complains** | | | |
| Memory impairment | 49 (96.07%) | 70 (95.89%) | 0.958 |
| Behavioural changes | 33 (64.70%) | 11 (15.06%) | **0.000** |
| **Neuropsychiatric symptoms** | | | |
| Present | 50 (98.04%) | 19 (26.03%) | **0.000** |
| **Cognitive status** | | | |
| SCI (n = 39) | 11 (28.20%) | 28 (71.80%) | **0.048** |
| MCI (n = 85) | 40 (47.06%) | 45 (52.94%) | |
| **MCI Subtype** | | | |

*(Continued)*

**Table 1.** (Continued)

| Variables | MBI (n = 51) | Non-MBI (n = 73) | p-value |
|---|---|---|---|
| | Frequency (%) | Frequency (%) | |
| Amnestic MCI | 31 (77.5%) | 41 (91.11%) | **0.030** |
| Non-amnestic MCI | 9 (22.5%) | 4 (8.89%) | |
| **GDS** | | | |
| Median (range) | 4 (0–13) | 3 (0–9) | **0.001** |
| **NPI-Q Score** | | | |
| Median (Range) | 4 (3–7) | 0 (0–5) | **0.000** |
| **MBI Checklist** | | | |
| Median (Range) | 7 (1–26) | 0 (0–15) | **0.000** |

Abbreviations: MBI: Mild behavioural impairment; SCI: Subjective cognitive impairment; MCI: Mild cognitive impairment; GDS: Geriatric depression scale; NPI-Q Neuropsychiatric inventory questionnaire

considered at least six months or more NPS symptoms to define positive. We found that among the subjects with NPS, 72.46% met the criteria for MBI, which was lower in comparison to 90.3% and 95.43% reported in previous studies [7]. The prevalence rate of MBI was significantly (p = 0.048) higher in subjects with MCI(48.9%) as compared to (28.20%) SCI. Previous studies show similar findings for MCI[11,12]. But our study was one of the few to compare the prevalence with subjects with SCI. None of these previous studies found a statistically significant difference in the prevalence of MBI between MCI and non-MCI.A study which aimed to see the progressive changes in neuropsychological performance in subjects with MBI and without significant cognitive impairment[13] reported that MBI was associated with a faster decline in working memory and attention. On enquiring the subjects and their caregiver about the presence of personality changes 44 (35.48%) responded affirmatively, out of which two-thirds had MBI.

The socio-demographic parameters were comparable between MBI and non-MBI subjects except for marital status. There was significant difference in marital status between MBI and non-MBI group (91% vs 76%, p = 0.024), which is a unique finding of this study. A study which examined marital quality among spouses of subjects with MCI showed that various behaviours exhibited as a part of cognitive impairment were frequent and were distressing to spousal caregivers[14]. The finding in our study could reflect an increased identification of behavioural changes by a spouse when compared to a sibling or other caregivers.

Impulse dyscontrol, which involved two-thirds of the subjects, was the most commonly involved domain amongst the NPS. This was followed by decreased motivation (60.78%), emotional dysregulation (54.90%), social inappropriateness (21.57%) and abnormal perception (3.92%).

**Table 2. Frequency of MBI domains.**

| MBI Domains | Frequency | Percentage (%) |
|---|---|---|
| Decreased motivation | 31 | 60.78 |
| Emotional dysregulation | 28 | 54.90 |
| Impulse dyscontrol | 35 | 68.63 |
| Social inappropriateness | 11 | 21.57 |
| Abnormal perception | 2 | 3.92 |

Abbreviations: MBI: Mild behavioural impairment

**Table 3. Geriatric syndromes and MBI.**

| CGA | MBI | Without MBI | p-value |
|---|---|---|---|
| Vision | 24 (47.06%) | 31 (42.47%) | 0.612 |
| Hearing | 16 (31.37%) | 23 (31.51%) | 0.987 |
| Risk of fall | 21 (41.18%) | 21 (28.77%) | 0.151 |
| Mobility score | | | |
| Mean ± SD [95% CI] | 13.25 ± 2.28 [12.61–13.89] | 12.80 ± 2.011 [12.33–13.27] | 0.252 |
| Urinary incontinence | 17 (33.33%) | 7 (9.49%) | **0.001** |
| Nutrition | 8 (15.69%) | 6 (8.22%) | 0.196 |
| Memory | 26 (50.98%) | 31 (42.47%) | 0.349 |
| Depression | 26 (50.98%) | 20 (27.40%) | **0.007** |

Abbreviation: MBI: Mild behavioural impairment

Previous studies[7,10] indicated affective dysregulation as the most common domain. The higher incidence of impulse dyscontrol in our study could be due to higher reporting of irritability and aggression by relatives and higher proportion of male, seconded by other study which reported male predominance [15]. As the caregivers may consider impulse dyscontrol as hormonal dysregulation or normal ageing changes, it may not be brought to medical attention early in the course. We also found that affective dysregulation and decreased motivation domains were more likely to be seen in subjects with MCI. Previous studies suggested that anxiety[16] and apathy[17] predict a future decline in cognition. In keeping with the conceptualisation of NPS as makers of neurodegenerative disease, it is speculated that depression and anxiety may be manifestations of executive dysfunction. However, we did not find any statistically significant association between individual MBI domains and MCI.

Visual impairment[18], hearing loss[19], low gait speed[20], are few geriatric syndromes (GS), known to be risk factors for dementia as per previous literature. This study found a statistically significant association between urinary incontinence(p = 0.001) and MBI, mirroring the studies which showed an association between UI and depression[21] and anxiety[22].Urinary incontinence leads to distressing symptoms in older individuals[23] and is associated with dementia and frailty. But there are no studies which report association between UI and predementia states. This study is the first to report such an association, though the causal association cannot be determined.

**Table 4. Association with comorbidities.**

| Comorbidities | MBI (n = 51) (%) | Without MBI (n = 73) (%) | p-value |
|---|---|---|---|
| No Co morbidities | 11 (21.57%) | 23 (31.51%) | **0.037** |
| One co morbidity | 10 (19.61%) | 24 (32.88%) | |
| Two or more co morbidities | 30 (58.82%) | 26 (35.62%) | |
| Hypertension | 33 (64.71%) | 41 (56.16%) | 0.340 |
| Diabetes | 23 (45.10%) | 15 (20.55%) | **0.004** |
| Hypothyroidism | 7 (13.73%) | 5 (6.85%) | 0.203 |
| CAD | 6 (11.76%) | 5 (6.85%) | 0.343 |
| CVA | 5 (9.80%) | 5 (6.85%) | 0.552 |
| COPD | 5 (9.80%) | 8 (10.96%) | 0.836 |
| Hyperlipidemia | 3 (5.88%) | 4 (5.48%) | 0.924 |

Abbreviation: MBI: Mild behavioural impairment, CAD: Coronary artery disease; CVA: Cerebrovascular accident; COPD: Chronic obstructive pulmonary disease.

Our study also revealed that more than half the subjects with MBI had multimorbidity (two or more comorbidities) when compared to one-third of the subjects without MBI. This association was statistically significant(0.037). Study by Mortby ME et al. did not report any association between the number of comorbidities and MBI[24]. Among the comorbidities, the prevalence of hypertension, diabetes, hypothyroidism, coronary artery disease, cerebrovascular accident and hyperlipidemia were higher in subjects with MBI. However, diabetes was significantly high(p = 0.004) in MBI. A study on subjects with Alzheimer's disease which aimed to examine the association between the presence of neuropsychiatric symptoms and vascular risk factors such as hypertension, diabetes and hypercholesterolemia[25] found that these comorbidities were associated with a higher prevalence of specific NPS. The NPS which were specifically associated with diabetes were delusion, hallucinations, agitation, irritability and sleep disturbances. There are numerous mechanisms which try to explain the cognitive decline in individuals with diabetes mellitus, these include insulin dysregulation, pro-inflammatory pathways, vascular risk factors, oxidative stress and lipoprotein receptors[26], which are similar to the causal pathway between NPS and cognitive decline[27]. Our study is the first to report an association between diabetes and MBI.

This study has strengths and limitations like any other cross sectional study. We have enrolled patients referred to a tertiary care memory clinic for cognitive concerns, which caters a heterogeneous population. Thus, the data can be considered as representative sample. For the purpose of diagnosing Mild Cognitive Impairment, CDR was used, which can identify mild but significant cognitive impairment. To make the diagnosis of MBI more accurately, all the four ISTAART-AA diagnosis criteria was used. It is also the first study on MBI from Indian sub-continent.

As it was a cross-sectional study, it is not possible to make a conclusion about changes in the prevalence of MBI over time. The risk of progression to cognitive impairment could not be determined.

## Conclusion

In a population attending memory clinic, with the diagnosis of SCI and MCI, we determined the prevalence of MBI using the ISTAART-MBI criteria. The prevalence of MBI was high, and significantly more prevalent in subjects with MCI, especially with amnestic MCI subgroup. Impulse dyscontrol was the most frequent MBI domain involved. These finding stressed upon the importance of assessing neuropsychological symptoms in subjects without dementia attending memory clinic. Multimorbidity, diabetes and urinary incontinence were significantly associated with MBI. So optimal management of multimorbidity, especially the cardiovascular morbidity (HTN, DM) might be potential preventive strategies and need further prospective study in large sample.

## Supporting information

**S1 Fig. Comorbidities in MBI and non-MBI subjects.** Bar diagram showing proportion of comorbidities in subjects with MBI and without MBI. Abbreviation: CAD: Coronary artery disease; CVA: Cerebrovascular accident; COPD: Chronic obstructive airway disease.
(EPS)

**S1 File. De-identified data file.**
(XLSX)

## Acknowledgments

This study was carried out at the Memory Clinic of the Department of Geriatric Medicine, All India Institute of Medical Sciences, New Delhi. Special acknowledgment goes to the team of Memory Clinic, which includes geriatricians (Dr Gaurav Desai, Dr Vijay Kumar, Dr Saroj), neuropsychologist (Dr Swati), nurses, physiotherapists and dietitian.

## Author Contributions

**Conceptualization:** Abhijith Rajaram Rao, Prasun Chatterjee, Aparajit Ballav Dey.

**Data curation:** Abhijith Rajaram Rao.

**Formal analysis:** Prasun Chatterjee, S. N. Dwivedi.

**Investigation:** Abhijith Rajaram Rao, Meenal Thakral.

**Methodology:** Abhijith Rajaram Rao, Meenal Thakral, S. N. Dwivedi.

**Project administration:** Prasun Chatterjee, Aparajit Ballav Dey.

**Resources:** Aparajit Ballav Dey.

**Writing – original draft:** Abhijith Rajaram Rao, Prasun Chatterjee, Aparajit Ballav Dey.

**Writing – review & editing:** Abhijith Rajaram Rao, Prasun Chatterjee, Meenal Thakral, Aparajit Ballav Dey.

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
