## [Decision Letter · Decision Letter 0]

6 Apr 2020

PONE-D-20-02291

Behavioural issues in late life may be the precursor of dementia- A cross sectional evidence from memory clinic of AIIMS, India

PLOS ONE

Dear Dr. Chatterjee,

Thank you for submitting your manuscript to PLOS ONE. After careful consideration, we feel that it has merit but does not fully meet PLOS ONE’s publication criteria as it currently stands. Therefore, we invite you to submit a revised version of the manuscript that addresses the points raised during the review process. We would appreciate receiving your revised manuscript by April 30, 2020. To enhance the reproducibility of your results, we recommend that if applicable you deposit your laboratory protocols in protocols.io, where a protocol can be assigned its own identifier (DOI) such that it can be cited independently in the future. For instructions see: http://journals.plos.org/plosone/s/submission-guidelines#loc-laboratory-protocols

We look forward to receiving your revised manuscript.

Kind regards,

Hemachandra Reddy

Academic Editor

PLOS ONE

Reviewers' comments:

Reviewer's Responses to Questions

**Comments to the Author**

1. Is the manuscript technically sound, and do the data support the conclusions?

Reviewer #1: Yes

Reviewer #2: Yes

2. Has the statistical analysis been performed appropriately and rigorously? 

Reviewer #1: Yes

Reviewer #2: Yes

3. Have the authors made all data underlying the findings in their manuscript fully available?

Reviewer #1: Yes

Reviewer #2: Yes

4. Is the manuscript presented in an intelligible fashion and written in standard English?

Reviewer #1: Yes

Reviewer #2: Yes

5. Review Comments to the Author

Reviewer #1: This is a clinical cross sectional study, that takes into consideration a syndromic term, MBI. In the cross sectional ascertainment the authors finds a high percentage of subjects met met MBI criteria, and impulse dyscontrol was very high. Seems a little strange but I suppose it is what it is.

Reviewer #2: 1. Please include the ethical statement in the manuscript under the materials and method section.

2. Statistical analysis could have been much better.

3. Result section: The authors just mentioned the table for the results, but there is no descriptive for them. Please elaborate on it correctly.

4. There are no proper footnotes for the table.

5. Have you done Apo-E analysis for those subjects involved in the present study?

6. The ethnic background of the population is missing in the table. Are subjects from only Delhi or nearby cities and states? Please clarify.

7. One or two small figures would help general readers to understand the findings.

8. The discussion could have been much better, and there are such essential references are missing.

6. PLOS authors have the option to publish the peer review history of their article (what does this mean?). If published, this will include your full peer review and any attached files.

Reviewer #1: No

Reviewer #2: No

---

## [Author Response · Author response to Decision Letter 0]

11 May 2020

Response to Reviewer #2:

1. Written informed consent was taken from participants. Ethical statement has been added under Materials and methods in the manuscript

3. Results section has been described in detail as advised

4. Footnotes have been added

5. No, Apo-E analysis has not been done of the study subjects

6. The tertiary care centre caters to the population of Delhi as well as nearby cities and states. This statement has also been mentioned in Materials and methods in the manuscript

7. One figure showing the comorbidities between the groups has been added

8. Discussion has been modified

---

## [Editor Report · Decision Letter 1]

28 May 2020

Behavioural issues in late life may be the precursor of dementia- A cross sectional evidence from memory clinic of AIIMS, India

PONE-D-20-02291R1

Dear Dr. Chatterjee,

We are pleased to inform you that your manuscript has been judged scientifically suitable for publication and will be formally accepted for publication once it complies with all outstanding technical requirements.

With kind regards,

Hemachandra Reddy

Academic Editor

PLOS ONE
---

## [Editor Report · Acceptance letter]

1 Jun 2020

PONE-D-20-02291R1 

Behavioural issues in late life may be the precursor of dementia- A cross sectional evidence from memory clinic of AIIMS, India 

Dear Dr. Chatterjee:

I am pleased to inform you that your manuscript has been deemed suitable for publication in PLOS ONE. Congratulations! Your manuscript is now with our production department. 

With kind regards,

on behalf of

Dr. Hemachandra Reddy 

Academic Editor

PLOS ONE